# Genetic Diversity of Selected Rice Genotypes under Water Stress Conditions

**DOI:** 10.3390/plants10010027

**Published:** 2020-12-24

**Authors:** Mahmoud M. Gaballah, Azza M. Metwally, Milan Skalicky, Mohamed M. Hassan, Marian Brestic, Ayman EL Sabagh, Aysam M. Fayed

**Affiliations:** 1Rice Research and Training Center (RRTC), Field Crops Research Institute, Agricultural Research Center, Kafr El-Sheikh 33717, Egypt; Mahmoudgab@yahoo.com; 2Molecular Biology Department, Genetic Engineering and Biotechnology Institute, University of Sadat City, Sadat City 32897, Egypt; azzaher2000@yahoo.com (A.M.M.); aisamge2004@yahoo.com (A.M.F.); 3Department of Botany and Plant Physiology, Faculty of Agrobiology, Food, and Natural Resources, Czech University of Life Sciences Prague, Kamycka 129, 165 00 Prague, Czech Republic; skalicky@af.czu.cz (M.S.); marian.brestic@uniag.sk (M.B.); 4Department of Biology, College of Science, Taif University, P.O. Box 11099, Taif 21944, Saudi Arabia; m.khyate@tu.edu.sa; 5Department of Plant Physiology, Slovak University of Agriculture, Nitra, Tr. A. Hlinku 2, 949 01 Nitra, Slovakia; 6Department of Agronomy, Faculty of Agriculture, Kafrelsheikh University, Kafr El-Sheikh 33516, Egypt

**Keywords:** rice, drought stress, genetic diversity, SSR markers, dendrogram

## Abstract

Drought is the most challenging abiotic stress for rice production in the world. Thus, developing new rice genotype tolerance to water scarcity is one of the best strategies to achieve and maximize high yield potential with water savings. The study aims to characterize 16 rice genotypes for grain and agronomic parameters under normal and drought stress conditions, and genetic differentiation, by determining specific DNA markers related to drought tolerance using Simple Sequence Repeats (SSR) markers and grouping cultivars, establishing their genetic relationship for different traits. The experiment was conducted under irrigated (normal) and water stress conditions. Mean squares due to genotype × environment interactions were highly significant for major traits. For the number of panicles/plants, the genotypes Giza179, IET1444, Hybrid1, and Hybrid2 showed the maximum mean values. The required sterility percentage values were produced by genotypes IET1444, Giza178, Hybrid2, and Giza179, while, Sakha101, Giza179, Hybrid1, and Hybrid2 achieved the highest values of grain yield/plant. The genotypes Giza178, Giza179, Hybrid1, and Hybrid2, produced maximum values for water use efficiency. The effective number of alleles per locus ranged from 1.20 alleles to 3.0 alleles with an average of 1.28 alleles, and the He values for all SSR markers used varied from 0.94 to 1.00 with an average of 0.98. The polymorphic information content (PIC) values for the SSR were varied from 0.83 to 0.99, with an average of 0.95 along with a highly significant correlation between PIC values and the number of amplified alleles detected per locus. The highest similarity coefficient between Giza181 and Giza182 (Indica type) was observed and are susceptible to drought stress. High similarity percentage between the genotypes (japonica type; Sakha104 with Sakha102 and Sakha106 (0.45), Sakha101 with Sakha102 and Sakha106 (0.40), Sakha105 with Hybrid1 (0.40), Hybrid1 with Giza178 (0.40) and GZ1368-S-5-4 with Giza181 (0.40)) was also observed, which are also susceptible to drought stress. All genotypes are grouped into two major clusters in the dendrogram at 66% similarity based on Jaccard’s similarity index. The first cluster (A) was divided into two minor groups A1 and A2, in which A1 had two groups A1-1 and A1-2, containing drought-tolerant genotypes like IET1444, GZ1386-S-5-4 and Hybrid1. On the other hand, the A1-2 cluster divided into A1-2-1 containing Hybrid2 genotype and A1-2-2 containing Giza179 and Giza178 at coefficient 0.91, showing moderate tolerance to drought stress. The genotypes GZ1368-S-5-4, IET1444, Giza 178, and Giza179, could be included as appropriate materials for developing a drought-tolerant variety breeding program. Genetic diversity to grow new rice cultivars that combine drought tolerance with high grain yields is essential to maintaining food security.

## 1. Introduction

Rice is the most diversified crop, which is grown under diverse ecological conditions. It is the staple food for more than 50% of the world’s population and is the world’s most important food in terms of a natural calorie source [1]. It occupies almost one-fifth of the total land area covered under cereals [2]. Due to the changing climate, the frequent occurrence of many extreme events contributes to different abiotic stresses, limiting the productivity of rice globally. Among them, drought is one of the most critical abiotic stresses that continually threatens the world’s food security [3]. The severity of the drought depends on many factors, such as the occurrence and distribution of rainfall, evaporative demands, and moisture-retaining capacity of the soils [4]. Therefore, it is imperative to find out the genotypes that can grow under water-scarce conditions to expand rice-growing areas in water-limited lands. It can be helpful to meet the challenge of ever-increasing global food demand [5]. Different rice varieties of a distinct genetic structure promise a future improvement of rice cultivars against drought stress [6]. Hence, the assessment of genetic diversity becomes important in establishing relationships among different cultivars [7]. The first step towards determining the magnitude of these risks is to evaluate the genetic diversity in improved rice genotypes as the success of a crop improvement program depends on the importance of genetic variability and how the desirable characters are heritable [8]. This identification of genotypes and their inter-relationships is important. The development of new biotechnological techniques provides increased support to evaluate genetic variation in both phenotypic and genotypic levels. The results derived from analyses of genetic diversity at the DNA level could be used for designing effective breeding programs aiming to broaden the genetic basis of commercially grown varieties.

Molecular marker technology is a powerful tool for determining genetic variation in rice varieties. In contrast to morphological traits, molecular markers can reveal large differences among genotypes at the DNA level, providing a more direct, reliable, and efficient tool for germplasm characterization, conservation, and management and untouched by environmental influence [9]. SSR markers can detect a high level of allelic diversity, and they have been extensively used to identify genetic variation among rice subspecies [10]. Simple sequence repeats (SSR) markers are efficient in detecting genetic polymorphisms and discriminating among genotypes from germplasms of various sources; even they can notice the finer level of variation among closely related breeding materials within the same variety [11]. Several quantitative trait loci (QTLs) in rice with consistent effects on grain yield under water-limited conditions were reported [12]. Among them, DTY1.1, located in chromosome 1 of the rice genome, was identified from rice variety N22 and successfully transferred to susceptible genotypes line IR64 and MTU1010 [13]. Besides this, two other major-effect QTLs viz. DTY3.1 and DTY2.1 were also identified, which explains about 30% and 15% of the phenotypic variance, respectively [14]. Later, Shamshudin et al. [15] reported another two QTLs, DTY2.2 ad DTY12.1, for reproductive stage drought tolerance in rice. However, as mentioned earlier, all of these QTLs are derived from stable grain yield under drought conditions. Because of the low heritability of grain yield under drought stress, selection for secondary traits was more effective than grain yield traits. Due to the lack of a useful trait selection index related to drought tolerance, it is essential to find a molecular marker associated with drought tolerance in rice [16]. Many SSR markers have been reported to be linked to drought tolerance traits in rice, such as yield under drought [13,17,18]. Although several investigations have researched rice germplasm characterization and diversity analysis, variability studies of the common landraces and cultivars grown are limited. Therefore, the study was conducted based on three main goals; (i) morphological characterization of the genotypes for grain and agronomic parameters under normal and drought stress conditions; (ii) genetic differentiation of 16 rice cultivars by determination of specific DNA markers related to drought tolerance using SSR markers and; (iii) grouping of cultivars according to their genotypes and subsequent decision their genetic relationship for different traits.

## 2. Materials and Methods

### 2.1. Experiment Site

The field experiment was conducted in Rice Research and Training Center Farm, Sakha Research Station, Agricultural Research Center, Egypt, in consecutive two rice growing seasons during 2018 and 2019 to investigate morphological traits and genetic diversity of 16 rice genotypes. The soil’s physical and chemical properties in the Sakha Research Station in 2018 and 2019 years are illustrated in Table 1.

### 2.2. Treatments and Design

The sixteen rice genotypes origin, pedigree, salience, and feature are shown in Table 2. All selected rice materials were grown under full irrigated (normal) and water stress conditions (flush irrigation every twelve days and exposed after fifteen days from transplanting) in a randomized complete block design with three replications.

### 2.3. Experimental Procedures

Seeds of all cultivars were sown in a nursery on 5 May and transplanted into the main field after 30 days in both years (2018 and 2019). A single seedling of each genotype was transplanted in 5 rows having 20 by 20 cm space (between rows and within row distance). Data were recorded from 10 randomly selected plants from each genotype. For characterization of root structure, large iron cylinders of 20 cm diameters and 60 cm height were used. They were buried inside the soil with a hammer, dug out with a spade, and pulled out using hooks. The roots were separated from the soil by thorough washing in a special washing facility. After taking the quantitative data, the shoot was separated from the root using a sharp knife and dried in an oven at 70 °C for five days. Root length (cm) was measured by the length of the root from the base of the plant to the tip of the longest root, root volume was determined by measuring the volume of water displaced by the plant root system (mm^3^), root thickness, the average diameter (mM) of the tip portion (about 1 cm from the tip) of three random secondary roots at the middle position of the root/plant, the number of roots/plant were estimated by the account roots at the maximum tillering stage and root: shoot ratio the ratio of the root dry weight (g) to the shoot dry weight (g) at maximum tillering stage was measured. Days to heading was recorded after flowering by the daily count of panicle exertion. The physiological maturity dates were recorded when 80% grains turn into golden yellow color. The leaf rolling scores were estimated by visual estimation, and the susceptible varieties and lines first started the rolling symptoms in the morning. Highly sensitive lines did not unroll at early morning hours and were recorded based on methods proposed by (De Data et al. 1988). The flag leaf area (cm^2^) was measured using a leaf area meter (LI-3100 (LI-COR Inc., Lincoln, NE, USA), plant height (cm) was measured in (cm), from the soil surface to the tip of the tallest panicle of each plant, relative water content (%) was measured using the formula (Fw − Dw) × 100/(Tw − Dw) where, Fw is fresh leaf weight, Dw is leaf dry weight, and Tw is turgid leaf weight. A number of panicles/plant were recorded at harvest by counting the number of panicles/plant, 100-grain weight (g) was recorded as the weight of 100 random chosen filled grains/plant, sterility percentage (%) was calculated by the divided number of unfilled spikelets/panicle on a number of total spikelets/panicle, grain yield/plant (g) was recorded by collecting the filled grains from all the tillers in a single plant and their weight recorded, water use efficiency was calculated as economic yield/total water consumed during the crop growth period.

All cultural practices were applied as recommended. Nitrogenous fertilizer was used in three splits as top dressing; phosphorus and potash were applied in full dose at sowing. Insect and weed control was used as and when required.

### 2.4. Statistical Analysis

A combined analysis of variance for the two years was carried out for the yield and yield components. Phenotypic correlation between yield and yield-related traits was done following Steel et al. [19], and the data were analyzed using the Co-State software program.

### 2.5. Genomic DNA Extraction

Ten seeds of each advanced genotype were placed into a Petri dish with filter paper soaked in distilled water for germination under aseptic conditions. Then, germinated seeds were grown into labeled pots. Genomic DNA was extracted from the healthy portion of young leaves harvested from 21 days old seedlings. DNA isolation was carried out using a mini preparation modified CTAB (cetyltrimethylammonium bromide) method, which did not require liquid nitrogen, and only a minimal amount of tissue samples were needed [20]. Leaf tissues were cut into small pieces, homogenized, and digested with extraction buffer (1 M Tris, 0.5 M Na 2EDTA, 5 M NaCl, and distilled Hybrid 2O, pH 8.0) and 20% SDS. Following incubation of leaf extracts for 10 min at 65 °C in a water-bath, 100 µL of 5M NaCl was added and mixed well by gentle inversion. Then 100 µL 10 × CTAB was added and again incubated for 10 min at 65 °C in a water-bath. After that, 900 µL of a mixture of chloroform and isoamyl alcohol (24:1) was added and centrifuged for 8 min at 11,000 rpm in a microcentrifuge. Then, 500 µL of the upper aqueous layer was separated, and 600 µL of ice-cold isopropanol was added to it, mixed, and centrifuged for 12 min at 13,200 rpm. A small pellet was visible, and the supernatant was decanted. The pellet was then washed with 200 µL cold 70% ethanol and centrifuged at 13,200 rpm for 12 min. After removing ethanol followed by air drying, the DNA pellets were re-suspended into 100 µL of 1× TE buffer and dissolved the pellet by warming in a 65 °C water bath for up to 1 h (with frequent mixing or flicking the tube with finger). Then the pellet was stored at −20 °C in an ultra-freezer. The quality of DNA was estimated by agarose gel (0.8%) electrophoresis and visualized with UV light.

### 2.6. SSR Markers and PCR Amplification

Ten SSR markers related to drought tolerance traits/QTLs were used. The sequences of primer pairs are found on the Web database (http://www.gramene.org). Primers’ names, repeat motifs, chromosome number, and related trait/QTL are shown in Table 3. PCR amplification reactions were done in 10 µL reaction mixtures, containing 50 ng/µL of template DNA, 0.5 µL of each forward and reverse primer, 5 µL of PCR master mix (Ferments), and 3 µL dd H_2_O. Thermal cycler was used with the following PCR profile: an initial denaturation step at 94 °C for 5 min, followed by 35 cycles of denaturation at 94 °C for 1 min, annealing at 55 °C for 30 s, and primer elongation at 72 °C for 1 min and then a final extension at 72 °C for 5 min. Amplified products were stored at −20 °C until further use.

### 2.7. Electrophoretic Separation and Visualization of Amplified Products

Five µL of PCR amplified product was loaded into each well of 3% agarose gel supplemented with ethidium bromide. The TAE 1× was used as a running buffer, and a 50 bp DNA ladder (0.5 μg/μL, ferments) was used to estimate the molecular size of the amplified fragments. Electrophoresis was conducted at 60 Volts for 2 h. Gels were then visualized and photographed using a Biometra gel documentation unit (Bio-Doc, Biometra, Germany).

### 2.8. SSR Data Analysis

The amplified SSR DNA bands representing different alleles were scored as different genotypes. For each marker, allelic bands were compared against a 100 bp DNA ladder. Then, fragment data was converted into the binary encoded allelic data to apply the multivariate analyses. Genetic distance, the ratios of shared DNA bands, and genetic similarities were estimated from the allele binary formatted data set using Nei and Li’s coefficient [26]. Genetic distance was calculated as follows:GDn = 1 − [2N11/(2N11 + N10 + N01)]
where N (1,1) is the number of loci having bands present in both accession, N (1,-) is the number of loci having a band present in the first accession, N (-, 1) is the number of loci having a band present in the second accession.

The accessions were clustered based on the matrix of genetic similarities using the unweighted pair group method with arithmetic averages (UPGMA). Polymorphic information content (PIC) values were calculated for each microsatellite based on the allelic frequency detected in the accessions studied using this formula.

Where, Pij is the frequency of the j-th allele for the i-th marker, and summation extends over n alleles. Polymorphic loci were defined as those whose most frequent allele had a frequency of less than 0.95.

Genetic diversity of the entries/populations (based on a set of measured molecular data) was estimated using diversity parameters other than PIC [27]. These are calculated as follows: percentage of polymorphic loci (PPL):P = (k/n) × 100%
where k is the number of polymorphic loci, n is the total number of loci investigated. The average number of alleles per locus (A):A = Σ Ai/n
where Ai is the number of alleles at the i-th locus and n is the total number of loci investigated. The average number of alleles per polymorphic locus (Ap):Ap =ΣApi/np
where, Api is the number of alleles at a certain polymorphic locus, np is the total number of polymorphic loci investigated.

Percentage of polymorphic alleles (PPA)
PPA = (Σ Api/Σ Ai) × 100%

The similarity matrix using [26] genetic distance for SSR characterization was also used for principal coordinate analysis (PCoA) with the Dcenter, Eigen, Output, and Mxplot subprograms in NTSYS-PC.

## 3. Results and Discussion

The analyses of variance in Table 4; Table 5 showed that the mean squares due to years were significant for major studied traits, except for days to heading, which would indicate overall vast differences among the genotypes studied annually. Abdallah et al. [28] observed the ordinary analysis of variance showed highly significant differences among environments, genotypes, and environments × genotype interaction for root and shoot traits in both treatments (normal and drought). The variation due to interaction between year and variety was not significant for all measures [29]. Mean squares due to environments were highly significant for all traits studied, indicating that all environments showed significant differences. Mean squares due to genotype × environment interactions were highly significant for all traits except, root thickness, flag leaf area, plant height, relative water content, number of panicles/plant, 100-grain weight, sterility percentage, and grain yield/plant, which indicated that the tested genotypes varied from the environment to environment and ranked differently from the normal condition. Raman et al. [30] recorded that the variance analysis for grain yield indicated a highly significant genotype x degree of stress severity interaction. Mean squares due to genotype × year interactions were significant for all traits studied, except root length, root volume, root thickness, number of roots/plants, root: shoot ratio, and leaf rolling.

Some genotypes surpassed the others once the mean squares of genotypes were highly significant than the interaction G × Y mean squares and identified the most superior genotypes. Genotype × environment × year mean squares were not significant for all the studied traits, except, days to heading and leaf rolling, indicating that each genotype’s performance in one environment will be changed from one year to another. The significant differences among rice genotypes in this investigation revealed genetic variability in the studied material and provided an excellent yield improvement opportunity. Grain yield and other characteristics exhibited stability across the seasons as significant genotype × environment interaction, which indicated the differences among genotypes were apparent (Table 4; Table 5). This research shows that further improvement through the selection of all studied characteristics could be effective. Genotype characteristics that confer an advantage in some water stress environments may prove useless or may even be a liability in other environments. This is reflected in the large G × E interactions in drought trials and the difficulty of identifying drought-tolerant check cultivars Zhang et al. [31].

### 3.1. Performance Across Environments

The ordinary analysis of variance indicated highly significant differences among genotypes for all traits studied in Table 4; Table 5 in the combined data. The studied genotypes’ mean performances at the combined data over environments are presented in Table 6, Table 7 and Table 8. For root length, the genotypes showing high values were IET1444, Sakha101, Sakha106, and Hybrid 2 (26.39, 25.88, 25.73, and 25.01cm, respectively), while the lowest values were obtained from Sakha102, Giza178, Sakha104, and Sakha103 (21.94, 21.42, 19.83 and 18.71cm, respectively). The genotypes Giza178, Hybrid 1, IET1444, and Sakha101 gave the superior values for root volume, 64.06, 57.66, 54.84, and 52.28 mm^3^, respectively; otherwise, the genotypes Giza177, Giza182, Sakha105, and Sakha103 gave the lowest one 103, 35.00, 29.88, 27.27, and 16.50 mm^3^, respectively. Concerning the root thickness, the genotypes E. Yasmine, Hybrid 1, IET1444, and Hybrid 2 had increased values of 1.08, 1.08, 1.07, and 0.97 mM. On the other hand, Giza178, Sakha102, Sakha104, and Giza182 resulted in the decreased values of 0.58, 0.56, 0.46, and 0.41 mM, respectively (Table 6).

Regarding the number of roots/plant, Sakha101, IET1444, Sakha104, Sakha106 produced the greatest number 272.65, 270.14, 266.86, and 248.20, respectively; meanwhile, the lowest number was found to be with Sakha103, Giza182, E. Yasmine, Giza181, recorded 169.13, 159.39, 157.18, and 135.81, respectively. The genotypes Giza181, Giza179, IET1444, and Giza178 recorded the major values 1.22, 1.13, 1.11, and 1.09 for Sakha103, 0.57, 0.55, 0.48, and 0.42, respectively. The genotype Giza178 produced the highest values for root volume and number of roots/plant. Hybrid 1 has superior values for the root volume. Root thickness and IET1444 have major root volume and root: shoot ratio under drought stress compared to other genotypes, indicating these genotypes can avoid water stress and increase the ability to absorb water from the soil. Moreover, efforts to increase yield under drought conditions also focused on improving secondary traits such as root architecture (root length, root volume, root thickness, number of roots, and root: shoot ratio). Rice genotypes that can maintain water status through adapted root systems come under the drought avoidance mechanism category. These genotypes can minimize the yield losses caused by drought [32]. Rice genotypes that avoid drought usually have deep, coarse roots with a high ability for branching and soil penetration and a higher root to shoot ratio [33]. Gaballah [34] reported the rice genotypes Moroberekan, Giza 178, and Sakha104 had the highest values for root characters under water shortage. Abdallah et al. [16] found the genotypes GZ5121-5-2 and GZ1368-S-5-4 had thicker roots, higher root diameter, and higher root length density than those grown under normal conditions.

The genotypes such as Sakha 102, Sakha 103, Giza177, and Sakha 105 took the least number of days to head and have earliness values of 94.42, 94.33, 94.17, and 93.92 days, respectively; in contrast, it took significantly longer time for E. Yasmine, Giza 181, GZ1368-S-5-4 and IET1444 having the earliness values of 118.25, 114.08, 112.67 and 112.50 days, respectively. These differences among rice genotypes might be attributed to their genetic background. The opposite strategy was observed in other cultivars, which had a significant delay in maturity with drought. Heading delay is a typical drought response observed in rice (Gaballah and Abd Allah) [35], which is expected to confer a benefit in those environments where stress is temporary, if development and flowering resume after the stress are relieved. Gaballah [34] mentioned that Moroberakan, Giza178, and Sakha104 gave the highest values under normal and drought conditions and for days to heading.

The desirable mean values for leaf rolling were found to be with Hybrid 1, Hybrid 2, GZ1368-S-5-4, and IET1444 were 2.08, 2.17, 1.58, and 1.50, respectively, while, the undesirable mean values observed with ‘Giza177′, Giza178, Giza179 and ‘Giza181′were 3.21, 2.33, 2.17 and 3.67, respectively (Table 7). In this investigation, drought tolerance can be assessed by visual scoring based on leaf rolling. A smaller degree of leaf rolling indicates a greater degree of dehydration avoidance by the development of deep roots. Gaballah [34] mentioned that the drought after 12 days increased leaf rolling in rice genotypes.

Concerning the flag leaf area, which is an important functional factor for photosynthesis, assimilation, and transpiration along the experimental plant life recorded the highest values with significant differences by the genotypes, IET1444, Giza181, E. Yasmine and Sakha101 were 25.08, 21.50, 20.62, and 20.57 cm^2^, respectively. Otherwise, the lowest value was recorded with Sakha102 (14.27 cm^2^). Reduced soil moisture levels produced a lower leaf area, which might inhibit cell division under water-starved conditions. Zubaer et al. [36] mentioned that the highest leaf area was found at 100% field capacity (FC) of the soil in all the rice genotypes. The leaf area was reduced by reducing moisture levels, but the degree reduction was higher in Basmati (14.7 for 70% FC and 53.2% for 40% FC) cm. There were significant differences in the plant height among the studied rice genotypes, suggesting that the growth rates were different in these genotypes. With respect to plant height in Table 7, the most desirable mean values towards dwarfism were obtained in the genotypes Sakha101, Giza 179, Sakha105, and Giza 177, which were 85.43, 85.72, 86.60, and 87.13cm, respectively. Similarly, the tallest values were obtained in Sakha106, Sakha104, Sakha102, and IET1444, 96.13, 96.86, 100.45, and 104.37 cm, respectively. Rice cultivars have tolerance by assessing plant height reduction under drought stress conditions. Lafitte et al. [37] indicated that the low land stress reduced height by only 4 cm (3%), ranging from a 43 cm reduction to a 22 cm increase in height. Regarding relative water content, the genotypes Sakha104, IET1444, Giza179, and Sakha101 resulted in the maximum values 83.30, 83.72, 83.76, and 84.55%, respectively, while the minimum values were recorded in Sakha105, Sakha103, ‘Giza177′ and Giza181 were 53.61, 58.78, 61.24, and 66.87%, respectively (Table 7). Plant responses to tissue water potential determine their level of drought tolerance. The traits, such as leaf turgor (RWC) maintenance and leaf rolling, have been used as selection criteria in rice [38], due to rice cultivar’s ability to save water in the leaf tissue to overcome water shortage. In the present study, we could also find a similar mechanism of drought tolerance is operating in genotypes, such as Giza 179, Sakha 101, Sakha 104, and IET1444, which were able to maintain significantly higher RWC under drought condition. We found that these genotypes can be considered as tolerant to moderately tolerant rice genotypes for drought stress.

For a number of panicles/plant, the genotypes Giza179, IET1444, Hybrid 1, and Hybrid 2 recorded the highest mean values of 20.16, 20.18, 20.70, and 22.08, respectively. Otherwise, the lowest values 13.92, 14.81, 14.95, and 15.35 with the genotypes GZ1368-S-5-4, Sakha102, Sakha106, and Giza177, respectively (Table 8). The heaviest 100-grain weight were 3.00, 3.03, 3.14, and 3.19 g, achieved with Sakha102, Sakha106, Giza182, and E. Yasmine. Therefore, the genotypes GZ1368-S-5-4, Giza178, IET1444, and Giza181 gave the lightest values, 2.10, 2.25, 2.30, and 2.40 g, respectively. The desirable values for sterility percentage were confirmed with genotypes IET1444, Giza178, Hybrid 2 and Giza179, which were 9.58, 9.81, 10.05 and 10.28%, respectively, otherwise the genotypes Sakha106, Sakha104, Giza177, and Giza181 gave undesirable values 14.53, 15.75, 16.08, and 16.25%, respectively. Concerning grain yield/plant, the genotypes Sakha101, Giza179, Hybrid 1, and Hybrid 2 resulted in the greatest values 35.56, 37.51, 40.00, and 41.50 g/plant. On the other hand, the lowest values obtained with the genotypes IET1444, Giza181, Giza182, and E. Yasmine were 27.14, 28.08, 29.96, and 31.03 g/plant, respectively. Mukamuhirwa et al. [39] reported that the cultivar Intsindagirabigega was most tolerant to drought, while Zong geng was the most sensitive. The genotypes Giza178, Giza179, Hybrid 1, and Hybrid 2 gave maximum values 0.78, 0.84, 0.88, and 0.92 respectively, for water use efficiency, while the genotypes Giza181, IET1444, Sakha105, and Sakha103 recorded the minimum values were 0.62, 0.62, 0.66, and 0.67, respectively. The rice genotypes Giza 179, Hybrid 1and Hybrid 2 had higher values for a number of panicles/plants, 100-grain weight, grain yield/plant and water use efficiency under normal, drought stress, and their combined data. Gaballah and AbdAllah [35] mentioned that the water stress reduced plant height, induced leaf rolling in the susceptible rice genotypes. The reduction of grain yield, number of panicles/plants, 100-grain weight, and high sterility percentage resulted from water stress at flowering and ripening stages. Water stress during vegetative, panicle initiation, flowering has reduced grain yield/plant by 28%, 34%, and 40%, respectively. Drought mitigation, through the development of drought-tolerant varieties with higher yields suitable for water-limiting environments, will be the critical factor in improving stable rice production.

### 3.2. Number of Alleles and Allelic Diversity

The sixteen rice genotypes were used in the present study were subjected to DNA polymorphism screening and assessment using SSR markers, which offer excellent potential for generating large numbers of markers evenly distributed throughout the genome and have efficiently been used to give reliable and reproducible genetic markers. Ten SSR primer pairs related to drought tolerance with known map positions distributed in the rice genome were used to screen a set of sixteen selected indica, japonica, and tropical-japonica rice genotypes with different levels with mechanisms of drought.

Among 10 SSR markers, spread on seven chromosomes (1, 2, 4, 5, 6, 8, and 9) generated polymorphic alleles. Table 9 showed that a total number of 85 alleles were detected at the nine markers’ loci across the sixteen rice genotypes. The number of alleles per locus generated by each marker varied from 2 to 15 alleles, with an average of 8.5 alleles per locus. The effective number of alleles per locus ranged from 1.20 alleles to 3.0 alleles, with an average of 2.28 alleles. The highest number and the effective number of alleles per locus were observed for RM263 (3.0), RM289 (2.81), and RM242 (2.63). Similar results for a low number of alleles per locus were also obtained by [40] (3.33) and [41] (2.5). On the contrary, a high number of alleles per locus was obtained by [42] (8.57). There was a significant positive correlation between the number of alleles detected at a locus and the number of repeats within the targeted microsatellite DNA (r = 0.57 **). Thus, the larger the repeat number in the microsatellite DNA, the larger the number of alleles detected. Moreover, it was reported that the dinucleotide repeat motif (GA) displayed a high level of variation among the rice genotypes [24]. On the other hand, [40] reported no correlations between the number of alleles detected and the number of SSR repeats.

### 3.3. Gene Diversity

The gene diversity or heterozygosity (H_e_) of a locus is defined as the probability that an individual is heterozygous for the locus in the population [43]. Higher values of this measure tend to be more informative because there is more allelic variation. As shown in Table 9, the H_e_ values for all SSR markers used in this study varied from 0.94 to 1.00, with an average of 0.98. The findings were in agreement with the observation of [44,45,46]. The highest H_e_ value (1.00) was recorded for RM23 and 518. Meanwhile, the lowest H_e_ values (0.98) were achieved by RM518.

### 3.4. PIC Value

PIC value refers to the value of a marker for detecting polymorphism within a population, depending on the number of detectable alleles and the distribution of their frequency; thus, it provides an estimate of the discriminating power of the marker [47]. As shown in Table 3, the PIC values for the SSR used in this study varied from 0.83 to 0.99, with an average of 0.95. This result is consistent with Sajib et al. [40] who reported high variation in PIC values for all tested SSR loci (from 0.14 to 0.71 with an average of 0.48). Higher averages of PIC values were reported by Zeng et al. [48] (0.57) and Ram et al. [49] (0.707). The highest PIC values were observed for RM23 (0.99), RM518 (0.99), RM223 (0.98) and RM276 (0.98). A highly significant correlation coefficient was found between PIC values and the number of amplified alleles detected per locus (r = −0.75 **), as shown in Table 9. A significant correlation between PIC value and the effective number of alleles (r = 0.92 **) and a highly significant correlation was found between PIC and gene diversity (r = −0.92 **). Similar results were obtained by Kumar et al. [50].

Figure 1 shows the PCR amplified fragments produced by the highest polymorphic markers in the current study, RM23, RM518, RM223, and RM276. These markers revealed the highest PIC values and gave the same values 1.00 and the highest number of alleles ranging from 8 to 9 alleles per locus, suggesting that these markers could be used for molecular characterization of a large number of rice genotypes rather than mapping populations for drought tolerance. The results were similar to those obtained by [10,51,52].

### 3.5. Identified MAS Marker

Among ten polymorphic SSR markers, RM518 was able to divide the studied genotypes into seven groups depending on their drought tolerance potential. The first group showed the first allele with a molecular weight of 125.89 bp included the drought susceptible genotypes japonica type1 and 2. The second allele with a molecular weight of 172.37 bp appeared in the second group included the drought moderate indica–japonica genotype Sakha105, Hybrid 1, Giza178, Giza181, and Giza182. The third the allele molecular weight 163.038 has six genotypes IET1444, Sakha101, Sakha102, Sakha106, Sakha104, and Giza177. The fourth allele molecular weight 375.85 bp with on genotype GZ1368-S-5-4 indica japonica type and tolerant to drought stress. The fifth allele molecular weight was 169.65 with one genotype Hybrid 2 moderate tolerant to drought and the six-allele molecular weight 158.56 bp with on cultivar Sakha103 susceptible to drought stress. This result agreed with [53] who reported that RM472 was linked to maximum root length and root dry weight characters. This marker could be useful in MAS for these characters in rice. The results were similar to those obtained by [46,54].

### 3.6. Similarity

The maximum similarity coefficient (0.5) was recorded between Giza181 and Giza182, which are indica type and drought susceptible genotypes at the same time (Table 10). Moreover, a high similarity percentage was observed between the japonica type and susceptible to drought stress Sakha104 with Sakha102 and Sakha106 (0.45), Sakha101 with Sakha102 and Sakha106 (0.40), Sakha105 with Hybrid 1 (0.40), Hybrid 1 with Giza178 (0.40) and GZ1368-S-5-4with Giza181 (0.40). On the other hand, no similarity percentage values were observed between the genotypes such as Sakha103, Giza179, IET1444, Sakha102, and Sakha104 with Giza178, GZ1368-S-5-4, Giza181, Giza182, E. Yasmine and Hybrid 2. These results were in harmony with [2], who reported a low similarity coefficient between japonica type and indica type genotypes, and [55] reported a relatively high level of similarity between closely related genotypes. Moreover, the findings are similar to those observed by [17,18].

### 3.7. Cluster Analysis

The genetic relationships among rice genotypes are presented in a dendrogram based on informative microsatellite alleles (Figure 2). All genotypes are grouped into two major clusters in the dendrogram at 66% similarity based on Jaccard’s similarity index. Whereas Jaccard’s similarity measure similarity for the two sets of data, it ranges from 0% to 100%. The higher the percentage, the more similar the two populations. Although it’s easy to interpret, it is extremely sensitive to small sample sizes and may give erroneous results, especially with very small samples or data sets with missing observations. The first cluster (A) is divided into two minor groups A1 and A2. The A1 sub-cluster included two groups A1-1 and A1-2, of which A1-1 contains genotypes IET1444, GZ1386-S-5-4, and Hybrid1, which are drought stress-tolerant. On the other hand, the A1-2 cluster was further divided into A1-2-1 (having Hybrid2 genotype) and A1-2-2 (containing Giza179 and Giza178) at coefficient 0.91, represented moderate or drought stress.

The A2 cluster, also subdivided into A2-1, included Giza181 and Giza182 at coefficient 0.94, while the A2-2 cluster had Egyptian Yasmine genotype, which has coefficient 0.76 with cluster A2-1. The A2 cluster comprised of the sensitive genotypes and indica type. Similarly, the cluster B was divided into two minor groups B1 and B2. The cluster B1 included B1-1 and B1-2 at coefficient 0.65. The B1-1 divided into B1-1-1 and B1-1-2 at coefficient 0.72. In general, the B cluster japonica type, and all were sensitive to drought stress. El-Malky et al. [42] reported the ability of SSR markers to divide the varieties into two distinct groups, one included the indica varieties, and the other had the japonica varieties. Moreover, Zeng et al. [48] found that all genotypes grouped into two major branches in the dendrogram with less than 10% similarity based on Jaccard’s similarity index, one unit represents the subspecies, japonica rice, and another unit represents the subspecies, indica, or the hybrids between japonica rice and indica rice.

## 4. Conclusions

Genetic improvement for drought tolerance in rice can be achieved in this present study based on results obtained at phenotypic characterization. Extensive genetic diversity analyses were presented as valuable in selecting the truly promising drought-tolerant genotypes, which can be used to cross to development genotypes with increased water stress tolerance levels. The rice genotypes Giza179, IET1444, Hybrid1, and Hybrid2, achieved the greatest values for grain yield/plant and produced maximum values regarding water use efficiency. It could be summarized the genotypes GZ1368-S-5-4, IET1444, Giza 178, and Giza179 were suitable materials for developing drought breeding. Thus, this study’s results indicate that incorporating genetic analyses with phenotypic data is very important to accelerate breeding programs by selecting suitable genotypes to improve target traits and could help exclude genotypes with bad performance.

## Figures and Tables

**Figure 1 plants-10-00027-f001:**
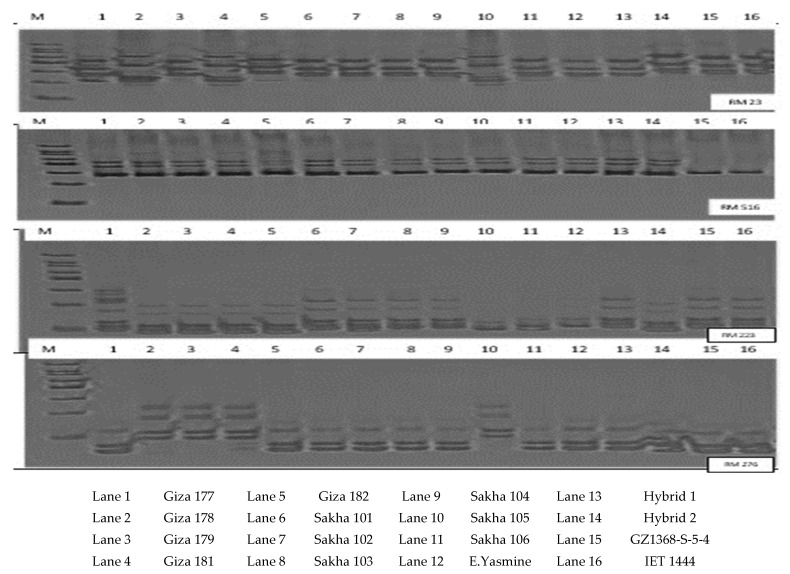
Agarose gel electrophoresis of PCR amplified fragments for the polymorphic SSR markers RM23, RM518, RM223, and RM276. M is a 100 bp DNA ladder.

**Figure 2 plants-10-00027-f002:**
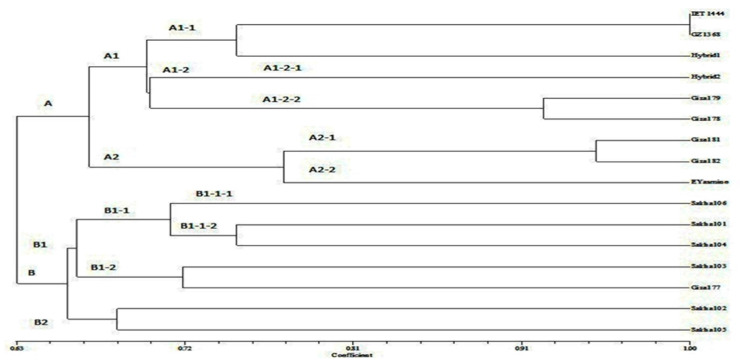
Dendrogram derived from unweighted pair group method with arithmetic averages (UPGMA) cluster analysis of sixteen rice genotypes based on Jaccard’s similarity coefficient using 10 SSR markers.

**Table 1 plants-10-00027-t001:** Physical and chemical properties of soil at Sakha Research Station in 2018 and 2019 years (the soil was collected before starting the field preparation in each season).

Soil Physical and Chemical Properties	Sakha, Kafr El-Sheikh	
	2018	2019
Clay %	55	55
Silt %	32.4	32.4
Sand %	12.6	12.6
Texture	Clayey	Clayey
Organic Matter	1.39	1.39
pH	8.1	8.2
Ec (Ds/m)	3.30	3.33
Total N (ppm)	512	518
Available P (ppm)	15.09	16.03
Co_3_^−^	-	-
Hco_3_^−^	5.55	5.56
Mg^++^	4.3	5
Na^+^	1.88	1.69
K^+^	16	16
Fe^++^	4.55	4.55
Mn^++^	3.1	3.5

**Table 2 plants-10-00027-t002:** Origin, pedigree, salience, and feature of sixteen rice genotypes.

Genotype	Pedigree	Salience and Feature
Giza 177	Giza 171/Yomjo No. 1//PiNo.4	Japonica type—sensitive to drought—short stature—early duration—resistance to blast
Giza 178	Giza175/Milyang 49	Indica/Japonica type, medium maturing, semi-dwarf, resistant to blast, medium grain, tolerant to drought, and high yield
Giza179	GZ6296/GZ1368	Indica/Japonica type—moderate to drought—short stature—early duration—resistant to blast
Giza181	(lR1626-203)	Indica type—sensitive to drought—short stature—late duration—resistance to blast
Giza182	Giza181/IR39422-161-1-3-1/Giza181	Indica type—sensitive to drought—short stature—early duration—resistant to blast
Sakha101	Giza 176/Milyang	Japonica type—sensitive to drought—short stature—long duration—sensitive to blast
Sakha 102	GZ4096-7-1/GZ4120-2-5-2 (Giza 177)	Japonica type—sensitive to drought—short stature, early duration—resistance to blast
Sakha 103	Giza177/Suweon349	Japonica type—sensitive to drought—short stature—early duration—resistance to blast
Sakha 104	GZ4096-8-1/GZ4100-9-1	Japonica type—sensitive to drought—long stature, moderate duration—sensitive to blast
Sakha 105	GZ5581-46-3/GZ4316-7-1-1	Japonica type—sensitive to drought—short stature—early duration—resistant to blast
Sakha 106	Giza176/Milyang79	Japonica type—sensitive to drought—long stature—early duration—resistant to blast
Hybrid 1	IR6962SA/Giza178	Indica type—moderate to drought—short stature—moderate
Hybrid 2	IR6962SA/Giza179	Indica type—moderate to drought—short stature—moderate duration—resistant to blast
E. Yasmine	Introduction	Indica type—moderate to drought—short stature late duration—resistant to blast
GZ 1368-S-5-4	IR 1615-31/BG 94-2349	Indica type—moderate to drought—short stature moderate duration—resistant to blast
IET1444	TN 1/CO 29	Indica type—tolerant to drought—short stature moderate duration—resistant to blast

**Table 3 plants-10-00027-t003:** Genetic details of SSR markers were used in the study.

Primer	Chromo-Some Number	Repeat Motif	Allele Size Range	No of Amplified Alleles	Effective Number of Alleles	Common Alleles	Gene Diversity	Polymorphic Information Content	Related Traits or QTLs	References
RM23	1	(GA)15	76.109—172.029	13.00	2.41	0.18	1.00	0.99	RSR, PH, DTH	[21]
RM164	5	(GT)16TT (GT)4	222.307—255.184	8.00	2.18	0.27	0.99	0.96	FGN, TSW, DTH	[22]
RM223	8	G11(GA)16	149.686—206.429	9.00	2.25	0.25	0.99	0.98	PH	[21]
RM242	9	(CT)25	127.35—130.068	7.00	2.63	0.37	0.98	0.95	RT, RL, SW	[21]
RM246	1	(CT)26	205.964—272.973	4.00	2.00	0.5	0.97	0.91	RL, SN, SSP	[21]
RM263	2	(CT)20	334.846—353.507	8.00	3.00	0.37	0.99	0.97	PH, RDW, RFW	[23]
RM272	1	(CT)34	144.029—146.487	2.00	1.20	0.6	0.94	0.83	DTH, SP, HI,	[24]
RM276	6	(GA)9	127.065—316.725	10.00	2.00	0.2	0.99	0.98	PL, LL, LW, SN	[24]
RM289	5	(TC)15	101.834—144.426	9.00	2.81	0.31	0.99	0.97	RV, PL	[24]
RM518	4	(AG)8A3 (GA)33	125.898—277.807	15.00	2.37	0.16	1.00	0.99	DTM, PH	[25]

RSR (root to shoot ratio), PH (Plant height), DTH (Days to heading), FGN (Filled grain number), TSW (1000-seed weight), RT (Root Thickness), RL (Root Length), SW (Seed weight), SN (Seed number), SSP (seed sterility percentage), RDW (root dry weight, RFW (root fresh weight), SP (Seed Percentage), HI (Harvest index), PL (panicle length), LL (Leaf Length), LW (leaf width), SN (Spikelet number), RV (Root volume), DTM (days to maturity).

**Table 4 plants-10-00027-t004:** Analysis of variance for growth characteristics under normal and drought conditions.

Source of Variance	df	Root Length (cm)	Root Volume (mm^3^)	Root Thickness (mm)	Number of Roots/Plant	Root:Shoot Ratio	Days to Heading (Day)	Leaf Rolling
Blocks	2	53.5051 **	2.584	0.0051	23.507	0.0157	1.02	1.51 **
Year (Y)	1	63.19 **	178.33 **	0.055 *	4555.10 **	0.0845 **	11.02 **	0.09
Env. (E)	1	842.23 **	15528.61 **	4.931 **	677517.6**	16.96 **	927.52 **	487.82 **
Gen. (G)	15	56.27 **	1801.26 **	0.465 **	22490.53 **	0.702 **	859.53 **	6.78 **
Y × E	1	1.256	15.019 *	0.0016	407.19 **	0.030	0.001	0.34
Y × G	15	0.033	1.2263	0.0008	14.055	0.007	4.75 **	0.34
E × G	15	15.06 **	178.02 **	0.0131	13297.93 **	0.384 **	4.30 **	4.68 **
Y × E × G	15	0.23	0.508	0.0002	7.30	0.004	1.91 **	0.46 *
Error	126	0.577	2.9094	0.00138	19.2036	0.0001	0.70 **	0.26
Total	191							

*, ** significant and high significant at probability 0.05 and 0.01, respectively. Env. (Environment), Gen. (Genotypes).

**Table 5 plants-10-00027-t005:** Analysis of variance for grain yield and related traits under normal and drought in 2018 and 2019 rice growing seasons.

Source of Variance	df	Flag Leaf Area (cm^2^)	Plant Height (cm)	Relative Water Content (%)	Number of Panicles/Plants	100-Grain Weight (g)	Sterility Percentage %	Grain Yield/Plant (g)	Water Use Efficiency
Blocks	2	1.04	4.48	1.138	9.48 **	0.19 **	2.83	0.07	0.00116
Year (Y)	1	1.51	6.27	314.75 **	5.27 *	0.31 **	5.60 *	3.69	0.0013
Env. (E)	1	1113.6 **	34978.5 **	18680.1 **	1566.37 **	2.34 **	6394.08 **	14,121 **	0.004 **
Variety (V)	15	84.41 **	369.10 **	2834.5 **	75.30 **	1.25 **	57.32 **	185.97 **	0.091 **
Y × E	1	3.52 **	0.01 **	11.746 *	2.80	0.00 **	1.84	0.04	0.00004
Y × V	15	5.24 **	4.42*	2.533	2.65 **	0.001	2.97 **	4.48 **	0.003 **
E × V	15	6.49 **	185.62 **	248.41 **	13.45 **	0.10 **	90.42 **	61.88 **	0.024 **
Y × E × V	15	1.57	1.99	0.6180	0.97	0.001	1.33	1.71	0.001 *
Error	126	0.94	2.50	2.0424	0.77	0.00	1.12	1.13	0.0006
Total	191								

*, ** significant and high significant at probability 0.05 and 0.01, respectively.

**Table 6 plants-10-00027-t006:** Performance of Growth characteristics under normal and drought conditions and their combined data.

Variety	Root Length (cm)			Root Volume (mm^3^)			Root Thickness (mm)			Number of Roots/Plant			Root:Shoot Ratio		
	d	n	C	d	n	C	d	n	C	d	n	C	d	n	C
Giza177	20.50	24.60	22.55	29.01	41.00	35.00	0.51	0.92	0.72	95.84	290.38	193.11	0.41	1.08	0.74
Giza178	19.27	23.58	21.42	51.25	76.88	64.06	0.44	0.72	0.58	197.83	281.57	239.70	0.62	1.56	1.09
Giza179	22.35	25.63	23.99	26.65	47.50	37.08	0.62	0.92	0.77	160.41	265.48	212.94	0.78	1.48	1.13
Giza181	20.30	26.65	23.47	26.96	46.13	36.54	0.41	0.82	0.62	117.88	153.75	135.81	0.60	1.85	1.22
Giza182	20.71	24.09	22.40	23.88	35.88	29.88	0.21	0.62	0.41	138.38	180.40	159.39	0.26	1.33	0.79
Sakha101	23.06	28.70	25.88	37.93	66.63	52.28	0.51	0.92	0.72	185.53	359.78	272.65	0.54	0.59	0.57
Sakha102	19.27	24.60	21.94	27.37	43.56	35.47	0.41	0.72	0.56	133.76	299.61	216.69	0.49	0.99	0.74
Sakha103	16.91	20.50	18.71	9.94	23.06	16.50	0.51	0.80	0.66	133.25	205.00	169.13	0.31	0.53	0.42
Sakha104	19.68	19.99	19.83	34.13	51.25	42.69	0.31	0.62	0.46	174.25	359.47	266.86	0.47	0.94	0.71
Sakha105	18.35	27.16	22.76	21.22	33.31	27.27	0.51	0.92	0.72	78.41	302.38	190.39	0.45	0.66	0.55
Sakha106	23.78	27.68	25.73	41.00	51.25	46.13	0.51	0.92	0.72	184.50	311.91	248.20	0.38	0.93	0.66
E. Yasmine	23.58	26.14	24.86	40.69	61.50	51.10	0.92	1.23	1.08	115.00	164.37	157.18	0.45	0.51	0.48
Hybrid 1	20.19	26.65	23.42	51.25	64.06	57.66	0.92	1.23	1.08	127.50	231.96	179.73	0.38	1.40	0.89
Hybrid 2	21.83	28.19	25.01	23.88	61.50	42.69	0.82	1.13	0.97	126.59	241.59	184.09	0.48	0.98	0.73
GZ1368-S-5-4	21.53	22.55	22.04	37.62	48.69	43.15	0.51	0.92	0.72	183.99	276.03	230.01	0.50	0.99	0.75
IET1444	26.14	26.65	26.39	46.13	63.55	54.84	0.91	1.23	1.07	175.28	365.00	270.14	0.72	1.51	1.11
LSD0.05	1.18	1.23	0.61	2.61	3.37	1.38	0.16	0.21	0.1	7.7	7.71	3.54	0.025	0.25	0.09
LSD0.01	1.71	1.78	0.86	3.78	4.88	1.95	0.23	0.30	0.14	11.15	11.16	5.01	0.04	0.36	0.13

d, n, and c are drought, normal, and combined data, respectively.

**Table 7 plants-10-00027-t007:** Growth characteristics of genotype under normal, drought conditions, and their combined data.

Variety	Days to Heading (Days)			Leaf Rolling			Flag Leaf Area (cm^2^)			Plant Height (cm)			Relative Water Content (%)		
	d	n	c	d	n	c	d	n	c	d	n	c	d	n	C
Giza 177	92.33	96.00	94.17	5.25	1.17	3.21	17.10	21.25	19.18	74.85	99.42	87.13	41.97	80.50	61.24
Giza 178	104.17	108.00	106.08	3.33	1.33	2.33	15.00	19.42	17.21	79.37	106.92	93.14	78.47	85.10	81.79
Giza 179	92.67	96.33	94.50	3.33	1.00	2.17	15.00	23.23	19.12	74.20	97.23	85.72	82.41	85.10	83.76
Giza 181	111.17	117.00	114.08	5.83	1.50	3.67	19.67	23.33	21.50	82.83	98.75	90.79	59.20	74.53	66.87
Giza 182	92.67	96.33	94.50	5.83	1.50	3.67	15.00	21.00	18.00	78.53	99.15	88.84	77.68	83.20	80.44
Sakha 101	101.50	107.00	104.25	5.50	1.50	3.50	18.33	22.80	20.57	70.93	99.92	85.43	82.00	87.10	84.55
Sakha 102	92.67	96.17	94.42	5.33	1.17	3.25	11.97	16.57	14.27	82.13	118.77	100.45	76.88	88.40	82.64
Sakha 103	92.67	96.00	94.33	5.33	1.33	3.33	14.30	18.68	16.49	77.75	109.07	93.41	51.25	66.30	58.78
Sakha 104	98.67	104.50	101.58	5.09	1.67	3.38	16.53	18.33	17.43	85.40	108.32	96.86	82.10	84.50	83.30
Sakha 105	92.67	95.17	93.92	5.42	1.33	3.38	14.00	17.83	15.92	75.53	97.67	86.60	29.21	78.00	53.61
Sakha 106	92.83	96.17	94.50	5.42	1.17	3.29	13.35	17.83	15.59	81.98	110.28	96.13	71.80	86.53	79.16
E. Yasmine	115.33	121.17	118.25	5.33	1.33	3.33	17.75	23.48	20.62	78.07	110.15	94.11	71.69	80.50	76.09
Hybrid 1	102.67	106.67	104.67	3.17	1.00	2.08	15.93	19.80	17.87	76.67	101.23	88.95	69.86	88.40	79.13
Hybrid 2	100.83	105.67	103.25	3.17	1.17	2.17	15.67	20.33	18.00	74.42	100.20	87.31	62.36	80.50	71.43
GZ1368-S-5-4	110.50	114.83	112.67	2.17	1.00	1.58	16.87	23.58	20.23	79.33	97.92	88.63	65.56	82.80	74.18
IET1444	109.17	115.83	112.50	1.83	1.17	1.50	22.05	28.10	25.08	79.90	128.83	104.37	81.18	86.25	83.72
LSD0.05	1.40	1.21	0.68	0.87	0.75	0.41	1.31	1.95	0.78	3.46	2.14	1.28	1.22	1.54	1.16
LSD0.01	2.03	1.75	0.96	1.26	1.09	0.58	1.90	2.82	1.10	5.01	3.10	1.81	1.77	2.23	1.64

d, n, and c are drought, normal, and combined data, respectively.

**Table 8 plants-10-00027-t008:** Grain yield performance and related traits under normal, drought conditions and their combined data.

Variety	Number of Panicles/Plants			100-Grain Weight (g)			Sterility Percentage%			Grain Yield/Plant (g)			Water Use Efficiency		
	d	n	c	d	n	c	d	n	c	d	n	c	d	n	c
Giza 177	12.03	18.67	15.35	2.74	2.96	2.85	25.33	6.83	16.08	22.03	42.40	32.22	0.68	0.75	0.71
Giza 178	17.50	21.33	19.42	2.14	2.36	2.25	14.78	4.83	9.81	26.50	43.70	35.10	0.80	0.77	0.78
Giza 179	16.07	24.25	20.16	2.71	2.74	2.73	14.83	5.73	10.28	28.78	46.23	37.51	0.86	0.83	0.84
Giza 181	14.30	17.17	15.73	1.98	2.81	2.40	22.83	9.67	16.25	20.33	35.83	28.08	0.62	0.62	0.62
Giza 182	16.33	19.82	18.08	3.03	3.24	3.14	18.58	7.48	13.03	22.75	37.17	29.96	0.71	0.66	0.68
Sakha 101	13.07	20.23	16.65	2.73	3.05	2.89	19.92	6.78	13.35	25.17	45.95	35.56	0.77	0.80	0.78
Sakha 102	12.95	16.67	14.81	2.94	3.06	3.00	20.17	7.33	13.75	22.33	42.50	32.42	0.68	0.74	0.71
Sakha 103	14.40	19.00	16.70	2.52	2.58	2.55	20.17	7.10	13.63	20.62	41.43	31.03	0.62	0.73	0.67
Sakha 104	12.50	20.08	16.29	2.53	2.72	2.63	25.50	6.00	15.75	25.17	44.13	34.65	0.77	0.77	0.77
Sakha 105	13.33	18.33	15.83	2.69	2.84	2.77	20.97	6.03	13.50	20.58	42.33	31.46	0.59	0.74	0.66
Sakha 106	10.73	19.17	14.95	2.93	3.12	3.03	22.00	7.05	14.53	25.00	42.80	33.90	0.77	0.76	0.76
E. Yasmine	17.33	22.00	19.67	3.03	3.34	3.19	14.72	10.92	12.82	26.77	35.28	31.03	0.80	0.60	0.70
Hybrid 1	16.73	24.67	20.70	2.48	2.64	2.56	22.97	6.00	14.48	30.17	49.83	40.00	0.89	0.87	0.88
Hybrid 2	19.33	24.83	22.08	2.75	2.88	2.82	14.40	5.70	10.05	30.50	52.50	41.50	0.92	0.92	0.92
GZ1368-S-5-4	12.50	15.33	13.92	1.94	2.25	2.10	12.87	14.73	13.80	27.10	38.52	32.81	0.80	0.67	0.74
IET1444	15.70	24.67	20.18	2.26	2.34	2.30	13.00	6.17	9.58	23.33	30.95	27.14	0.71	0.53	0.62
LSD0.05	1.00	1.43	0.71	0.37	0.33	0.001	2.15	1.54	0.85	1.48	1.18	0.86	0.04	0.02	0.02
LSD0.01	1.45	2.07	1.00	0.54	0.48	0.00	3.11	2.23	1.20	2.14	1.71	1.22	0.06	0.03	0.03

d, n, and c are drought, normal, and combined data, respectively.

**Table 9 plants-10-00027-t009:** The correlation coefficient for polymorphic SSR markers.

	No of Amplified Alleles	Effective Number of Alleles	Common Alleles	Gene Diversity
Effective number of alleles	1			
Common alleles	0.57 **	1		
Gene diversity	−0.48 **	−0.96 **	1	
Polymorphic information content	0.75 **	0.92 **	-0.92 **	1

** is highly significant at probability 0.001.

**Table 10 plants-10-00027-t010:** Similarity coefficient among studied genotypes based on SSR markers.

Genotypes	Sakha103	Giza179	IET1444	Sakha101	Sakha102	Sakha106	Sakha104	Giza177	Sakha105	Hybrid 1	Giza178	GZ1368-S-5-4	Giza181	Giza182	E. Yasmine
Giza179	0.35														
IET1444	0.18	0.13													
Sakha101	0.14	0.18	0.21												
Sakha102	0.19	0.18	0.26	0.40											
Sakha106	0.14	0.13	0.32	0.40	0.40										
Sakha104	0.18	0.23	0.25	0.32	0.45	0.45									
Giza177	0.05	0.04	0.08	0.18	0.18	0.30	0.35								
Sakha105	0.04	0.04	0.04	0.12	0.12	0.08	0.16	0.30							
Hybrid 1	0.04	0.04	0.04	0.08	0.08	0.04	0.07	0.13	0.40						
Giza178	0.00	0.00	0.00	0.04	0.04	0.04	0.00	0.04	0.17	0.40					
GZ1368-S-5-4	0.00	0.00	0.00	0.04	0.00	0.04	0.00	0.04	0.12	0.21	0.21				
Giza181	0.00	0.00	0.00	0.04	0.00	0.04	0.00	0.04	0.13	0.17	0.17	0.40			
Giza182	0.00	0.00	0.00	0.04	0.00	0.04	0.00	0.04	0.17	0.12	0.12	0.21	0.50		
E. Yasmine	0.00	0.00	0.00	0.00	0.00	0.00	0.00	0.04	0.13	0.18	0.18	0.17	0.25	0.30	
Hybrid 2	0.00	0.00	0.00	0.00	0.00	0.00	0.00	0.04	0.07	0.12	0.12	0.15	0.12	0.21	0.29

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
