# Peer review of "Genetic Diversity of Selected Rice Genotypes under Water Stress Conditions"

_plants, 2020, doi:10.3390/plants10010027_

Round 1

Reviewer 1 Report

The paper by Gaballah et al is scientifically weak. Some sections are very confusing and totally lacking a scientifically valid discussion. So,  for this reason, I cannot accept it for publication in the present form. Below some points to improve:

Materials and Methods

the authors should give more information about genetic of the varieties used in the present work.

Please in Table 1 the authors should report correct formula of the elements;

386-416. “Experimental procedures”: this section has to be completely rewritten because it is very confusing and does not reflect scientific standards;

Please in Table 2 the authors should insert footnotes, such as RSR, DTH and so on!

  1. Please rewrite this section! There are many errors in the text, such as PIC formula.

Authors should follow an order in the numbering of tables as well as the references used!

Results and discussion

109-111. this sentence is very confuse – please, rewrite it!

Tables 5 and 6 need an adequate statistic support, in order to facilitate the reader.

  1. “10 SSR markers…covering the whole rice genome”, this sentence it is scientifically risky.

Table 8 is completely wrong.

318. are the authors of this size (375.85) sure? I believe there might be some error

Figure 1 - please, improve it.

342. dendogram or dendrogram? Please, change it!

the authors should give more information reated to use of Jaccard’s similarity

Finally, in the paper the discussion section is almost absent.

Author Response

Dear Respect Editor,

Date: 8 December, 2020

To Editor of Plants;

 MDPI, St. Alban-Anlage 66, 4052 Basel, Switzerland

Sub: Submission of second revised version of the Manuscript: Ref. Ms. No. plants-1018865.

Thank you so much for considering our manuscript (Ref. Ms. No. plants-1018865) entitled " Genetic diversity of selected rice genotypes under water stress conditionand for sending your additional editorial as well as reviewers’ comments, which have allowed us to make considerable improvements to the manuscript. We are pleased to inform you that we have been able to address all of your and the reviewer’s requests. Please find our response in blue below the message and in the revised text in Red. If we missed any aspect by chance, please feel free to let us know.

We look forward to seeing our accepted manuscript published online in plants. We have fully revised the manuscript and all authors have seen and approved the final version for re-submission to plants.

Sincerest regards,

Ayman EL SABAGH (Corresponding author)

Dear reviewer

We would like to acknowledge your contribution explicitly. Thank you very much for your kind comments on the manuscript. Those comments are very helpful for improving the manuscript..  We have considered the comments and made correction. The corrections were highlighted in red in the revised version. The responses are as follows.

Material and methods

  1. The authors should give more information about genetic of the varieties used in the present work

Authors’ response:  The suggestion has been made and added table 2 and illustrated origin, pedigree, salience and feature of sixteen rice genotypes as well as more information about genetic of the varieties.

  1. Please in Table 1 the authors should report correct formula of the elements;

Authors’ response:  The correct form is in Table 1, it was corrected formula of the elements

  1. 386-416. “Experimental procedures”: this section has to be completely rewritten because it is very confusing and does not reflect scientific standards;

Authors’ response:  The suggestion has been made and 86-416. “Experimental procedures”: this section has modified to be more clear and followed scientific standards;

  1. Please in Table 2 the authors should insert footnotes, such as RSR, DTH and so on!

Authors’ response:  We have modified the text as per suggestion in Table 3, we inserted footnotes, for all abbreviations.

  1. 458 Please rewrite this section! There are many errors in the text, such as PIC formula.

Authors’ response:  The suggestion has been made and this section 458 has been rewritten.

  1. Authors should follow an order in the numbering of tables as well as the references used!

Authors’ response:  These references have been revised and the numbering of tables as well as the references has been done

Results and discussion

  1. 109-111. this sentence is very confuse – please, rewrite it!

Authors’ response:  The suggestion has been made  and 109-111. this sentence has been rewritten

  1. Tables 5 and 6 need an adequate statistic support, in order to facilitate the reader.

Authors’ response:  The suggestion has been made and Tables 5 and 6 in this form because we have normal and drought conditions and their combined data.

  1. “10 SSR markers…covering the whole rice genome”, this sentence it is scientifically risky.

Authors’ response:  The suggestion has been made and 10 SSR markers not covering the rice whole genome”, this sentence has changed.

  1. Table 8 is completely wrong.

Authors’ response:  Thanks for your good suggestions. We have revised with supervior and the table as per guided for us. This tables regarding correlation coefficient for polymorphic SSR markers, we reviewed the correlation analysis and it is presently in the right shape based on your advice.

  1. are the authors of this size (375.85) sure? I believe there might be some error.

Authors’ response:  The suggestion has been made and 318. Are the authors of this size (375.85) confirmed, based on the discussion with supervisor?  We believe there might be some error . Bands were detected on UV- trans-illuminator, photographed by Gel documentation system and according to analysis by Phoretix program 1D gel analysis software version 4.01.  And some SSR markers formed eight different band patterns (molecular weight 375 bp

  1. Figure 1 - please, improve it.

 Response: we have improved it based on your request and Figure 1 – it  has been improved.

  1. dendogram or dendrogram? Please, change it!

 Response: we have revised it based on your request and Dendrogram or dendrogram has been changed based on your request.

  1. the authors should give more information reated to use of Jaccard’s similarity

Already have been gained information related to use of Jaccard’s similarity

  Response: As suggested, we have revised it. Furthermore, we also revised the information for Jaccard’s similarity. Note: all changed according the reviewer comments do in MS. Manuscript and checked all manuscript data

  1. Finally, in the paper the discussion section is almost absent.

Response: we have improved it based on your request. Dear reviewer, do you mean the probability of the regression in this section.  We can clearly provide more discussion based on of your recommend.

Reviewer 2 Report

The Manuscript ID: plants-1018865 “Molecular characterization for rice genotypes under water stress condition” has many results, but it was not well prepared. The Abstract describes the experiments and the results, but does not address the study proposal and neither does the meaning of the results (conclusions, applications, utility ... ???).

According to the authors (Introduction), the objective of the study was the morphological characterization of the genotypes for grain and agronomic parameters under normal and drought stress conditions, the genetic differentiation of 16 rice cultivars by establishing specific DNA markers associated with drought tolerance using SSR markers.

The experiments and analysis to morphological characterization of the genotypes for grain and agronomic parameters under normal and drought stress conditions seems consistent. However, I don't understanded the analysis of the SSR loci. The 16 rice cultivars are not diploid ???

How were more than two alleles simultaneously detected in each locus of each cultivar (Figure 1) using agarose gel ??? In diploid organisms we can detect one or two alleles at each locus. SSR loci are codominant markers and it is possible to easily identify the homozygous and heterozygous plants. Analysis of SSR markers based on the presence or absence of bands, generating a binary data matrix of 1 and 0 for each marker is employed for polyploid species...

There are several words that have been digitized wrong. There are also inappropriate terms that were used (e.g. genetic make-up - line 185 ???).

In Results and Discussion: The results have been described and compared with the results of other studies, but frequently is not possible to understand whether the authors refer to the results of the present study or the study of other authors (e.g., lines 282-284).

The conclusions only summarized and emphasized the results beyond the theoretical expectations, but the practical and/or promising conclusions based on the results of the present study were not highlighted.

Rice culture is not my expertise. However I believe that the experiments and/or analysis of the SSR loci in the 16 rice cultivars should be remaked or rewritten in order to clarify the readers. The results of the experiments at the SSR loci are in disagreement with what is expected for diploid organisms. In addition, the analysis of the polymorphism in those loci is in disagreement with the codominant nature of the SSR markers.

In summary, the study has many results. However, the results from SSR markers need to be analyzed correctly. Results from morphological characterization of the genotypes for grain and agronomic parameters under normal and drought stress conditions seems consistent, but they need to be discussed in a practical/objective way, highlighting the effective contribution and the importance/utility of the results.

Author Response

Dear Respect Editor,

Date: 8 December, 2020

To Editor of Plants;

 MDPI, St. Alban-Anlage 66, 4052 Basel, Switzerland

Sub: Submission of second revised version of the Manuscript: Ref. Ms. No. plants-1018865

Thank you so much for considering our manuscript (Ref. Ms. No. plants-1018865) entitled " Genetic diversity of selected rice genotypes under water stress conditionand for sending your additional editorial as well as reviewers’ comments, which have allowed us to make considerable improvements to the manuscript. We are pleased to inform you that we have been able to address all of your and the reviewer’s requests. Please find our response in blue below the message and in the revised text in Red. If we missed any aspect by chance, please feel free to let us know.

We look forward to seeing our accepted manuscript published online in plants. We have fully revised the manuscript and all authors have seen and approved the final version for re-submission to plants.

Sincerest regards,

Ayman EL SABAGH (Corresponding author)

Dear reviewer

We would like to acknowledge your contribution explicitly. Thank you very much for your kind comments on the manuscript. Those comments are very helpful for improving the manuscript..  We have considered the comments and made correction. The corrections were highlighted in red in the revised version. The responses are as follows.

  1. The Manuscript ID: plants-1018865 “Molecular characterization for rice genotypes under water stress condition” has many results, but it was not well prepared. The Abstract describes the experiments and the results, but does not address the study proposal and neither does the meaning of the results (conclusions, applications, utility. ???).

Response: Thank you for you carefully comment, as suggested, abstract describes the experiments and the results, has changed and added address the study proposal and meaning of the results (conclusions, applications, utility).

  1. According to the authors (Introduction), the objective of the study was the morphological characterization of the genotypes for grain and agronomic parameters under normal and drought stress conditions, the genetic differentiation of 16 rice cultivars by establishing specific DNA markers associated with drought tolerance using SSR markers.

Response: we have revised this part as you mentioned, however, the analysis of the SSR loci. The 16 rice cultivars are diploid

  1. How were more than two alleles simultaneously detected in each locus of each cultivar (Figure 1) using agarose gel ??? In diploid organisms we can detect one or two alleles at each locus. SSR loci are codominant markers and it is possible to easily identify the homozygous and heterozygous plants. Analysis of SSR markers based on the presence or absence of bands, generating a binary data matrix of 1 and 0 for each marker is employed for polyploid species...

Response: The main aim of this section in this study was to, the agarose gel could detect more two alleles in each locus of each cultivar (Figure 1)

  1. There are several words that have been digitized wrong. There are also inappropriate terms that were used (e.g. genetic make-up - line 185???).

 Response: we have revised it., there are several words have been changed and corrected. e.g. genetic make-up - line 185 changed to genetic background.

  1. In Results and Discussion: The results have been described and compared with the results of other studies, but frequently is not possible to understand whether the authors refer to the results of the present study or the study of other authors (e.g., lines 282-284).

Response: Thank you so much for taking interest in this matter , the results of the present study or the study of other authors (e.g., lines 282-284), has been changed to be clearer.

  1. The conclusions only summarized and emphasized the results beyond the theoretical expectations, but the practical and/or promising conclusions based on the results of the present study were not highlighted.

Authors’ response:  The conclusion have been revised as per your suggestion and the conclusions only summarized and emphasized the results beyond the theoretical expectations, therefore the practical and/or promising conclusions based on the results of the present study has been highlighted.

  1. Rice culture is not my expertise. However I believe that the experiments and/or analysis of the SSR loci in the 16 rice cultivars should be remaked or rewritten in order to clarify the readers. The results of the experiments at the SSR loci are in disagreement with what is expected for diploid organisms. In addition, the analysis of the polymorphism in those loci is in disagreement with the codominant nature of the SSR markers.

Response: Thank you for your nice comment, the discussion section was included with results data and added discussion for part belongs to analysis of variance particularly (genotypes x Environment). Sir., In our limited experience, we am feeling sorry we can’t answer regarding experiments and analysis of SSR, because analysis of the SSR loci in the 16 rice cultivars was revised and remarked several times and confirmed by supervisors an professionals.

  1. In summary, the study has many results. However, the results from SSR markers need to be analyzed correctly. Results from morphological characterization of the genotypes for grain and agronomic parameters under normal and drought stress conditions seems consistent, but they need to be discussed in a practical/objective way, highlighting the effective contribution and the importance/utility of the results.

Response: Thank you so much for taking interest in this matter, we have done and re vised on all section.

Round 2

Reviewer 1 Report

In the present form, the paper does not follow the journal instructions such as references number and tables in the text and so on. Thus, I ask the authors to follow the indications given.

Author Response

Dear Respect Editor,

Date: 15 December, 2020

To Editor of Plants;

 MDPI, St. Alban-Anlage 66, 4052 Basel, Switzerland

Sub: Submission of second revised version of the Manuscript: Ref. Ms. No. plants-1018865.

Thank you so much for considering our manuscript (Ref. Ms. No. plants-1018865) entitled " Genetic diversity of selected rice genotypes under water stress conditionand for sending your additional editorial as well as reviewers’ comments, which have allowed us to make considerable improvements to the manuscript. We are pleased to inform you that we have been able to address all of your and the reviewer’s requests. Please find our response in Blue below the message and in the revised text in Blue. If we missed any aspect by chance, please feel free to let us know.

We look forward to seeing our accepted manuscript published online in plants. We have fully revised the manuscript and all authors have seen and approved the final version for re-submission to plants.

Sincerest regards,

Ayman EL SABAGH (Corresponding author)

Dear reviewer

A gain we would like to acknowledge your contribution explicitly. Thank you very much for your kind comments on the manuscript for second time. Those comments are very helpful for improving the manuscript. We have considered the comments and made correction. The corrections were highlighted in blue in the revised version.

In the present form, the paper does not follow the journal instructions such as references number and tables in the text and so on.

Authors’ response:  Thank you so much for sending your additional comments, which have allowed us to make an additional improvements of the manuscript. Thanks for your good suggestions. We have now formatted the MS as per guidelines for author and journal instructions.

Reviewer 2 Report

In the revised version of Manuscript ID plants 1018865, the analysis to morphological characterization of the genotypes for grain and agronomic parameters under normal and drought stress conditions were more properly explored and conclusives. However, I don't understanded how were more than two alleles simultaneously detected in each locus of each plant (Figure 1) using agarose gel... (???). In diploid organisms we can detect one or two alleles at each locus. SSR loci are codominant markers and it possible to easily and directly identify the homozygous and heterozygous plants. Analysis of SSR markers based on the presence or absence of bands, generating a binary data matrix of 1 and 0 for each marker is employed for polyploid species or to dominat markers. Genetic diversity analysis employed (represented) in the current study is usually adopted to molecular dominant markers.

The great advantage of using SSR markers is precisely the fact that these markers guarantee the identification of the number of homozygous and heterozygous plants in each cultivar. The genetic diversity seems that was not detected within each cultivar. I suspect that all the plants from each cultivar were mixed; is that it ??

The potential of the SSR loci was not explored in the present study.

If the plants from each cultivar were mixed, then is relevant to highlight that DNA was not extracted from each plant individually...

Author Response

Dear Respect Editor,

Date: 15 December, 2020

To Editor of Plants;

 MDPI, St. Alban-Anlage 66, 4052 Basel, Switzerland

Sub: Submission of second revised version of the Manuscript: Ref. Ms. No. plants-1018865.

Thank you so much for considering our manuscript (Ref. Ms. No. plants-1018865) entitled " Genetic diversity of selected rice genotypes under water stress conditionand for sending your additional editorial as well as reviewers’ comments, which have allowed us to make considerable improvements to the manuscript. We are pleased to inform you that we have been able to address all of your and the reviewer’s requests. Please find our response in blue below the message and in the revised text in Blue. If we missed any aspect by chance, please feel free to let us know.

We look forward to seeing our accepted manuscript published online in plants. We have fully revised the manuscript and all authors have seen and approved the final version for re-submission to plants.

Sincerest regards,

Ayman EL SABAGH (Corresponding author)

Dear reviewer

We would like to acknowledge your contribution explicitly for the second time. Thank you very much for your kind comments on the manuscript. Those comments are very helpful for improving the manuscript.  We have considered the comments. The corrections were highlighted in red in the revised version.  

Genotyping: Each band represents an allele having a specific size in base pairs.

Authors’ response:  Authors’ response:  Thank you so much for your informative question,  the Source of polymorphism: SSRs are one of several sequence variations named variable number of tandem repeats (VNTR), and are distinguished from the macrosatellites by the size of the core sequence (few nucleotides vs. several tens). Thus, the polymorphism is the number of the tandem repeats of a specific microsatellite at a specific locus.

Characteristics: SSRs are single locus markers and have a few to even more than ten alleles in each locus; thus, SSRs are highly polymorphic.

Authors’ response:  Thank you so much for your valuable comments, which have allowed us to make an additional improvements of the manuscript. SSRs are co-dominant markers so they can distinguish heterozygotes from homozygotes.

The main advantages of SSRs are their high level of polymorphism and their reliability. Authors’ response:  Thanks for your deeply explanations. Many studies have applied SSRs to various goals (Zhao and Kochert, 1993; Provan et al., 1998; Kim et al., 2002; Guo et al., 2006; Scascitelli et al., 2010).
